# Using the BWA (Bertaut-Warren-Averbach) Method to Optimize Crystalline Powders Such as LiFePO$_4$

**Aleksandr Bobyl [1,\*], Oleg Konkov [1], Mislimat Faradzheva [1] and Igor Kasatkin [2]**

[1] Ioffe Institute, Politekhnicheskaya Str. 26, St. Petersburg 194021, Russia; oleg.konkov@mail.ioffe.ru (O.K.); mislimat92@gmail.com (M.F.)

[2] Research Park, St. Petersburg State University, XRD Research Center, Universitetskaya nab. 7-9, St. Petersburg 199034, Russia; igor.kasatkin@spbu.ru

\* Correspondence: bobyl@theory.ioffe.ru

**Abstract:** The average sizes $\overline{L}_i$, and their dispersion $W_i$ along the $i$-th axis, of crystallites in powders are used to determine X-ray diffraction sizes, $D_{i\,XRD}$, averaged over crystallite columns within the BWA method. Numerical calculations have been carried out for an orthorhombic lattice of crystallites, such as LiFePO$_4$, NMC, having a Lamé's $g$-type superellipsoid shape. For lognormal distributions, the analytical expression for the normalized coefficient $K_n$ has been found: $K_n = D_{i\,XRD}/\overline{L}_i = (K_{g,0} + K_g W^2)$, where $K_{g,0}$ is a constant at W→0, $K_g$ is a constant depending on the $g$-type shape. The dependences of $D_{i\,XRD}$ are also calculated for normal distribution. A fairly simple equation can be obtained as a result of analytical transformations in the framework of experimentally validated approximations. However, a simpler way is to carry out numerical computer calculations with subsequent approximation of the calculated curves. Using the obtained analytical expressions to control technologies from nuclear fuel to cathode materials will improve the efficiency of flexible energy network, especially storage in autonomous and standby power plants.

**Keywords:** crystallites in powders; Lamé's shape; X-ray sizes; Bertaut–Warren–Averbach; normal distributions; lognormal distribution; energy storage; energy optimization

**MSC:** 49M05

## 1. Introduction

Scherrer began studying the shape of X-ray lines (LPA—line profile analysis) to determine the sizes of nanometer objects in 1918. The history is described in [1,2], and the latest results in [3]. In 1950, Bertaut [4], and Warren and Averbach [5] showed that the total diffraction intensity is the sum of individual diffractions from columns perpendicular to the planes that make up the crystallite. The magnitude of the size-induced broadening of diffraction reflections depends not only on the average sizes of crystallites, but also on the statistics of their distributions. The situation is described in detail in [4–9] and the following equation is assumed to hold:

$$D_{i\,XRD} \equiv \overline{D}_i(M_i) = \frac{1}{V}\int M_i\,dV = K_n \overline{L}_i \tag{1}$$

where $M_i$ is the length of the column along the $i$-th direction, $V$ is the crystallite volume, $K_n$ is the normalization coefficient of value $D_{i\,XRD}$ per crystallite linear size along the $i$-th axis. The result of averaging for a single ellipsoid, i.e., for a powder with a small dispersion, is obtained via integration [6–9]. In the case of crystalline powders with a non-orthorhombic lattice, it is necessary to take into account the angles between the axes [9].

Thus, the task of using the BWA method is reduced to determining the normalized coefficient $K_n$ by integrating (1) and summing over all crystallites, taking into account their shape, as well as the statistics of their distributions.

Based on TEM and SEM studies [2,7,10,11], various size distribution functions: lognormal, normal, gamma, and Poisson are normally used, more precisely, their one-dimensional projections onto some linear dimension. The first two of them can be used in their three-dimensional versions with pairwise correlations between sizes along the coordinate axes [12–14], while for the last two, such a possibility is not "transparent and easily accessible" [15–17]. In some exceptional cases, analytical approximations of distributions with a large dispersion of sizes may be useful [11]. This is probably also due to the fact that in some ranges of dispersion and average sizes, the lognormal distribution is visually close, for example, to the gamma distribution (see Supplementary Materials).

Currently, a large number of programs for processing diffractograms and determining $D_{i\,XRD}$ have been developed. First of all, it is necessary to take into account the instrumental contribution to diffraction line width [18,19], and then the general microstructure refinement of all peak profiles (WPPM—whole powder pattern modeling) procedure can be used [3,10,11]. The size–strain line broadening analysis can be combined with evaluating the crystallographic texture of various nanosized powders [20–22] in the Rietveld refinement software, in a technique known as Material Analysis Using Diffraction (MAUD) [23]. In TOPAS 3.0 v1 (Bruker) software, anisotropic crystallite size analysis is also implemented. The accuracy of the double-Voigt approach and the dimensional parameters obtained for triaxial ellipsoids, elliptic cylinders, and cuboid is discussed in [24,25].

Thus, there is a fairly reliable procedure for determining the X-ray diffraction sizes of anisotropic powders, in which their size distribution is assumed to be known a priori, or by inspecting histograms obtained by counting a small number of particles. In the absence of more complete information about the three-dimensionality of these distributions, the use of the BWA method in the form of Equation (1) loses its meaning due to a large error. The known value of the coefficient $K_n$ can reduce the error, which is very useful for more precise control of technology and predicting the target parameters of powders.

In this work, this problem is solved by the following numerical calculations: (1) the histograms of particle size distributions are simulated using 3D lognormal or normal functions in the form of a three-dimensional and $N$-bit matrix, (2) the matrix of averaged columns of crystallites is calculated with the shape of $g$-type superellipsoids and the sizes corresponding to the histogram bins, (3) the value of $\overline{D}_i(M_i)$, defined as the average column length of the whole powder sample along the $i$-th axis, is calculated via element-by-element multiplication of these two matrices and then summing the matrix elements along the $i$-th axis.

## 2. Calculation Model

### 2.1. Shape of Crystallites

Lamé's surface of $g$-type superellipsoids is described by the implicit equation

$$\left(\frac{2\,x_3}{L_3}\right)^g + \left(\frac{2\,x_2}{L_2}\right)^g + \left(\frac{2\,x_1}{L_1}\right)^g = 1 \tag{2}$$

superelipsoids belong to the superquadric family [26]

$$\left(\left|\frac{2\,x}{L_x}\right|^{\frac{2}{\epsilon_1}} + \left|\frac{2\,y}{L_y}\right|^{\frac{2}{\epsilon_1}}\right)^{\frac{\epsilon_1}{\epsilon_2}} + \left|\frac{2\,z}{L_z}\right|^{\frac{2}{\epsilon_1}} = 1 \tag{3}$$

Equation (2) is a special case of (3) with the ratio of powers: $\epsilon_1 = \epsilon_2 = 2/g$. The moduli signs are omitted in Equation (2), since the shapes of LiFePO$_4$ crystallites can be described even with only $g < 30$ [14,27] and their 0-th genus topology [28], i.e., excluding torus-type shapes.

Figure 1 shows anisotropic crystallites and columns $M_1$ along the 1st crystallographic direction. These forms are also used in the TOPAS Rietveld refinement [25]. Replacing $x_1$ in (2) with $\frac{M_1}{2}$, for the length $M_1$, we obtain:

$$M_1 = L_1\left(1 - \left((2\,x_2)/L_2\right)^g - \left((2\,x_3)/L_3\right)^g\right)^{\frac{1}{g}} \tag{4}$$

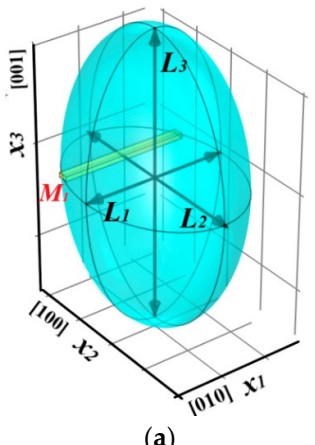
(**a**)

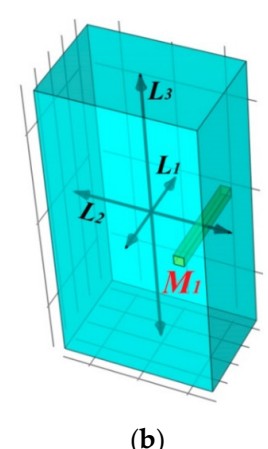
(**b**)

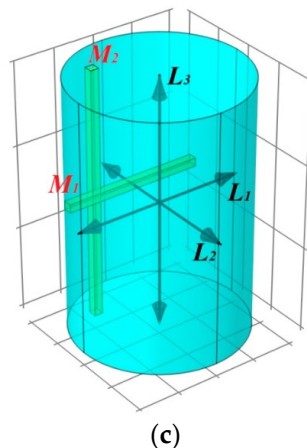
(**c**)

**Figure 1.** LiFePO$_4$ crystallite models: (**a**) ellipsoid, $g = 2$, $\epsilon_1 = \epsilon_2 = 1$, (**b**) cuboid $g = 30$, $g = 30$, $\epsilon_1 = \epsilon_2 = 0.2$, and (**c**) elliptical cylinder (bar), $\epsilon_1 = 1$, $\epsilon_2 = 0.04$ with dimensions $L_1$, $L_2$ and $L_3$ $M_1$ along the [010], [100], and [001] axes, respectively. $M_1$—lengths of columns with a cross section $dx_2 \cdot dx_3$, for (**c**) two directions are indicated.

For $\overline{D}_1(M_1)$ we obtain:

$$D_{1\,XRD} \equiv \overline{D}_1(M_1) = \frac{1}{V}\int M_1\,dV = \frac{1}{V}\iint M_1{}^2 dx_2 dx_3 = K_n \overline{L}_1 \tag{5}$$

where $V$ is the volume of the crystallite. The equality (5) is obtained for an ellipsoidal crystallite with orthogonal axes, and the coefficient $K_n = 3/4$ [7,9] results from normalizing to the average linear length of the crystallites. It will be seen below that in some cases, the dependence of $D_{1\,XRD}$ on only $\overline{L}_1$ is retained, while $K_n$ depends on the shape of the crystallites and the parameters of their size distribution. Variants with non-orthogonal axes are dealt with in detail in [9].

The double integration in (5) is carried out over the integration region $Reg$ of the central section of the ellipsoid perpendicular to the 1st axis,

$$Reg = \left(((2\,x_2)/L_2)^g + ((2\,x_3)/L_3)^g\right)^{1/g} \le 1 \tag{6}$$

The advantage of using Lamé's superellipsoids is the ability to determine their volume and cross-sectional area perpendicular to column $M_1$ [26,29], respectively, by using beta functions

$$V = \frac{1}{g^2}L_1 L_2 L_3 B\left(\frac{1}{g},\frac{2}{g}+1\right)B\left(\frac{1}{g},\frac{1}{g}+1\right) \tag{7}$$

$$S_{(010)} = \frac{1}{g}L_2 L_3 B\left(\frac{1}{g},\frac{1}{g}+1\right) \tag{8}$$

In particular, knowing the mass of the cathode powder, its density and the statistics of the size distribution of crystallites, with Equations (7) and (8), it is possible to determine the inner surface of the cathode and use it to improve the accuracy of determining the diffusion coefficient using the Randles–Sevcik equation [14], as well as for quantitative calculations of the rate of electrochemical reactions using the Butler–Volmer equation [27]. In this case, the shape of crystallites must be preliminarily determined by analyzing TEM and SEM images of crystallites either visually or using software processing methods [29–32].

### 2.2. Crystallite Statistics

In the study of the structural and electrochemical properties of LiFePO$_4$ powders, lognormal [33–39], normal [40–42], and Weibull [36,37] distributions were observed earlier. The latter was also used to describe carbon conglomerates of anode electrodes [39,43]. Gamma and lognormal distributions, as noted above [10,11], were used in the analysis

of specially tested $CeO_2$ powders. There are the following structural and technological justifications for their use:

1. In [44], it was shown that during the crushing of rocks, a lognormal distribution of products occurs, which is due to the multifactorial character of the crushing process itself.
2. In [45], the normal distribution arises as a result of aggregative growth and oriented attachment of nanocrystals.
3. In [43], the Weibull distribution results from the restructuring of the electrode morphology during battery usage.
4. Gamma is associated with exponential and normal distributions. In [46], the gamma distribution was observed at a constant rate of formation of nuclei near the supersaturation threshold of the solution and also during the formation of raindrops in [47,48].

In the Supplementary Materials, these 4 functions are used to construct the one-dimensional histograms of size (diameter) distributions for 8000 virtual spherical particles. They can be used at the first stage of visual selection of the experimental distribution type.

It was noted above that only the lognormal [13,14] and normal [12] distributions can be used to describe anisotropic particles, taking into account their pairwise covariances between the sizes. For the remaining two, they can also be adjusted to lognormal and normal with smaller errors compared to those arising when covariances are not taken into account.

In [49], marginal size distributions of crystallites in $LiFePO_4$ powders were determined along all three crystallographic axes, and in [14], the parameters of the 3-dimensional lognormal function $f_{LM}(\overline{L})$ were determined:

$$f_{LM}(\overline{L}) = \frac{1}{L_1 L_2 L_3 \sqrt{(2\pi)^3 \det\overline{\overline{\Lambda}}}} \exp\left[-\frac{1}{2}\left(\ln\overline{L} - \ln\overline{\overline{L}}\right)^T \overline{\overline{\Lambda}}^{-1}\left(\ln\overline{L} - \ln\overline{\overline{L}}\right)\right] \quad (9)$$

where $\overline{L} = \begin{bmatrix} L_1 \\ L_2 \\ L_3 \end{bmatrix}$ —crystallite sizes, $\overline{\overline{L}} = \begin{bmatrix} \overline{L_1} \\ \overline{L_2} \\ \overline{L_3} \end{bmatrix}$ —their mean values,

$\overline{\overline{\Lambda}} = \begin{bmatrix} W_1^2 & r_{12}W_1W_2 & r_{13}W_1W_3 \\ r_{21}W_2W_1 & W_2^2 & r_{23}W_2W_3 \\ r_{31}W_3W_1 & r_{32}W_3W_2 & W_3^2 \end{bmatrix}$ —matrix of correlation moments; the off-diagonal elements are covariances between the marginal distributions, for example, $Cov_{12} = Cov_{21} = r_{12}W_1W_2$ between the 1st and 2nd; $r_{12}$ is the correlation coefficient between them. The possible values of $r_{12}$ range from 0 to 1.

Below we also use the normal distribution $f_N(\overline{L})$ in the form [15]

$$f_N(\overline{L}) = \frac{1}{\sqrt{(2\pi)^3 \det\overline{\overline{\Lambda}}}} \exp\left[-2\left(\overline{L} - \overline{\overline{L}}\right)^T \overline{\overline{\Lambda}}^{-1}\left(\overline{L} - \overline{\overline{L}}\right)\right] \quad (10)$$

The most significant difference between (10) and (9) is the linear scale used in (10), and the dimension of the dispersion: in (9) the dispersion $W$ is dimensionless, and in (10), it has the dimension of length. In the calculations, it will be normalized to the average length along the $i$-th direction, which makes the calculations more general.

### 2.3. X-ray Size Calculation

According to (1), the X-ray diffraction size $\overline{D}_1(M_1)$ along the 1st axis is the average of the lengths of the columns $M_1$, as shown in Figure 1, over all powder crystallites. We obtain it by averaging over the $i$-th crystallite using integration (5) over the cross section (6) followed by averaging over all $N$ powder crystallites using function (9) or (10). To achieve this, we use the Mathematica 12.0 software for element-by-element matrix multiplication and their summation in the following form, for example, for the lognormal distribution:

$$\overline{\overline{D}}_1(M_1) = \textbf{Total}\left[\left(\overline{f_{LN}} \circ \overline{D}_1(M_1)\right)\right] \quad (11)$$

where $L_{in}$, $L_{jn}$, and $L_{kn}$ are the sizes of crystallites along the [010], [100], and [001] axes, respectively, the index $n$ varies from 1 to $N$, also for Normal $\overline{f_N}$. The crystallite volumes $V_N$ in Equation (6) will be represented by a 3-dimensional matrix (7), depending on the $g$-type of superellipsoid, in particular for an ellipsoid with $g = 2$, $V_N = \frac{\pi}{6} L_{in} * L_{jn} * L_{kn} \cdots \overline{f}$ in (8) stands for the discretization of function (9) or (10) in the form of a matrix normalized to 1, each element of which is the probability of superellipsoid presence with the corresponding size in the powder.

Thus, the task of Equation (11) is to determine the dependence of the coefficient $K_n$ in (5), on $g$–the type of superellipsoids and on the following parameters of the lognormal (9) or normal (10) distributions: $\overline{L}_i$–average sizes, $W_i$–dispersion, and $r_{ik}$–correlation coefficients of the matrix $\overline{\Lambda}$.

## 3. Results of Calculations and Discussion

In the presence of a large number of parameters, it is necessary to establish their hierarchy, evaluate their possible ranges, and also determine the number of bins (discreteness or dimension) of the histograms [50,51], which is equal to the matrix dimensions used in (11) in our case. It has been verified that the bin number should be at least 15–20 for a small discretization error, comparable to the errors of powder structural studies (see Supplementary Materials).

### 3.1. Lognormal Distribution

Parameter ranges. For high-quality LiFePO$_4$ powders, the parameters of the function $f_{LN}(\overline{L})$ fall in the following ranges: $\overline{L}_{1,2,3}$ ~40 ÷ 500 nm, $W_{1,2,3}$ ~0.3 ÷ 0.6, $r_{12,13,23}$~0.4 ÷ 0.7 [27].

To explain the calculation algorithm, the test results for 8000 crystallites and a 20-bin distribution histogram, through analogy with [27,49], are shown in Figure 2. Supplementary Materials contains similar figures and numerical values of 6-bit matrices, which were used to check and control the stages of calculations and to determine the discretization error. From Figure 2b, it can be seen that the histogram of X-ray diffraction size distribution shifts to the region of large sizes. This usually occurs with an increase in the dimensionality of the averages: from 1D to 3D. The linear dependence of $D_{1\ XRD}$ on $L_1$ exactly corresponds to the value of $K_n = \frac{3}{4}$ ($\frac{3}{4}$ of $\overline{L}_1$) in (5), Figure 2a (right scale). The sum of 270.41 nm is much larger than 240 nm, which suggests $K_n > \frac{3}{4}$, which is larger than expected for ellipsoidal crystallites with a small dispersion value $W$.

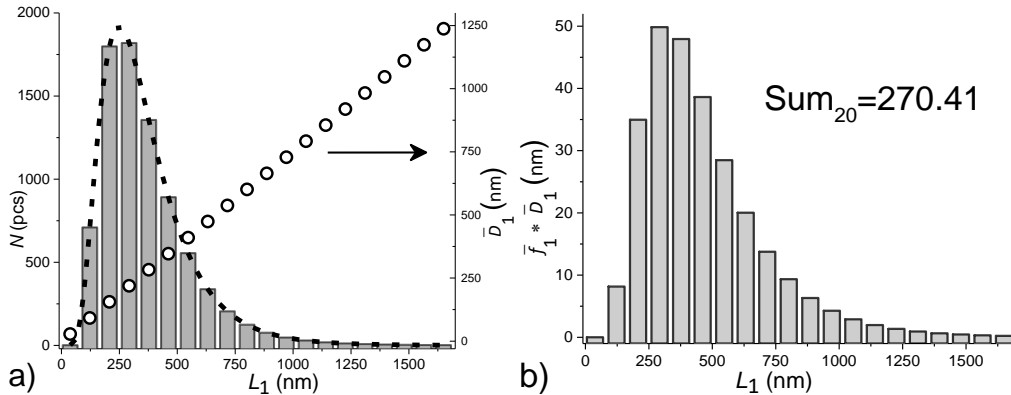

**Figure 2.** (**a**) Histogram simulation of ellipsoid crystallite size distribution ($g = 2$) along the [010] axis and its approximation by the dashed curve of the lognormal distribution with the parameters, $\overline{L}_1 = 320$ nm, $W_1 = 0.5$, $r_{12} = r_{13} = r_{23} = r$. Right axis averaging over the columns in each histogram bin. (**b**) Histogram of distributions of X-ray diffraction sizes and the total value of histogram bars.

Dependences of $K_n$ on correlation coefficients. The calculations illustrated in Figure 2 ignore the dependence of $K_n$ on the correlation coefficients $r_{ij}$ and on their anisotropy. Figure 3 shows the results of testing these assumptions and draws the following conclusions:

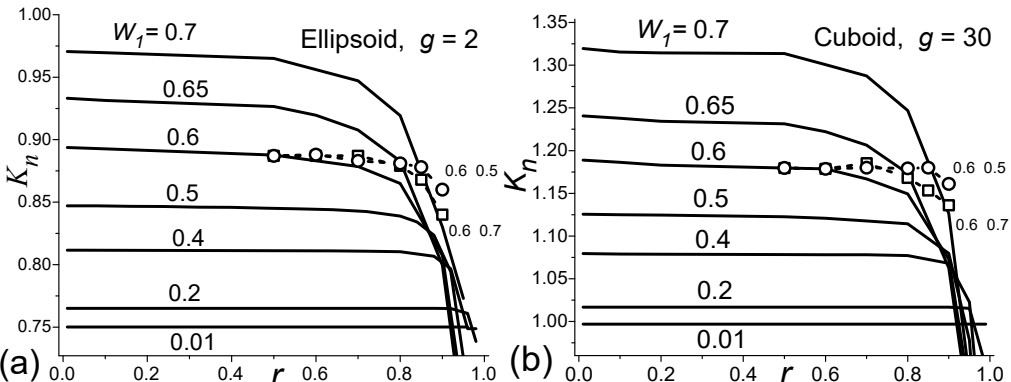

**Figure 3.** Dependences of $K_n$ on the correlation coefficients $r$ and dispersion $W_1$ of lognormal distribution for ellipsoids $g = 2$ (**a**) and close to cuboids for $g = 30$ (**b**) using 20-bit histograms. For $W_1 = 0.6$, calculations are shown using 30-bit histograms for $W_2 = W_3 = 0.5$ (o) and $W_2 = W_3 = 0.7$ (□).

1. After normalization $\overline{\overline{D}}_1(\overline{L}_1)$, the coefficients $K_n$ do not depend on the values of the average sizes $\overline{L}_1$ in the tested range from 80 to 600 nm, i.e., the normalization significantly expands the universality of the developed calculation procedure.
2. A weak dependence of $K_n$ on $r$ is observed at $r < 0.7$, i.e., in the range of values observed in high-quality LiFePO$_4$ samples. However, as shown in [27], this dependence is quite significant for the values of their electrochemical capacitances. Therefore, here and below, the calculations are performed for the value, $r_{12} = r_{13} = r_{23} = r = 0.5$.
3. The observed dependence at $r > 0.7$ is related to the histogram discreteness. Only in this region can the anisotropy of crystallite sizes along other crystallographic directions affect the calculation results. It also expands the universality of the developed calculation procedure.

Dependencies of $K_n$ on the dispersion $W_1$. Figure 4 shows the results of calculations that can already be directly used in the development of technology for electrode crystalline powders with a higher accuracy in determining their structural parameters compared to those previously used in [14,27] to determine the block structure of crystallites [49], to assess the potential of the developed technologies, and to optimize it [14]. From Figure 4, the following conclusions can be drawn:

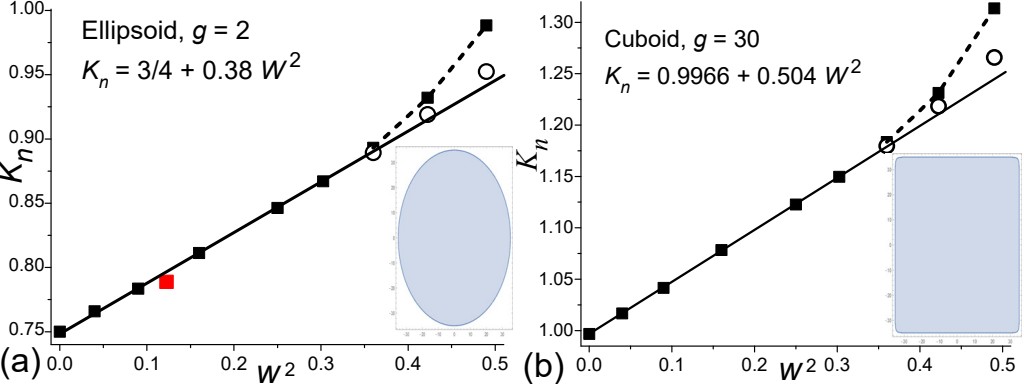

**Figure 4.** Dependences of $K_n$ on lognormal dispersions $W_1$ at $r = 0.5$ for ellipsoids (**a**) and cuboids (**b**) using 6-, 20- and 30-bit histograms: (■), (■) and (o), respectively. The red dot (■) is taken from SI. The straight line is an approximation by the equations shown in the figures.

1. For the lognormal crystallite size distribution the following analytical expression for $K_n$ is found

$$K_n = (K_{g,0} + K_g\, W^2) \tag{12}$$

   where $K_{g,0}$ is a constant at $W \to 0$, and $K_g$ is a constant, depending on the $g$-type shape. The values of these constants are collected in Table 1.

2. Equation (12) is valid up to the values of $W_1 \sim 0.6$. For larger values, it is necessary to increase the bit length of the histograms, which is possible with a $2 \div 3$-fold increase in the number of measured particles. On the other hand, the error of using 6-bit histograms with $2 \div 3$-fold smaller number of particles (up to $1000 \div 2000$) will be no more than 1%.

**Table 1.** Equation (12) constants depending on the $g$-type superellipsoid shape.

| $g$ | 2 | 4 | 6 | 8 | 10 | 30 | 50 |
|---|---|---|---|---|---|---|---|
| $K_{g,0}$ | 0.75 | 0.90 | 0.945 | 0.9654 | 0.9763 | 0.9966 | 0.9988 |
| $K_g$ | 0.38 | 0.46 | 0.483 | 0.4938 | 0.4993 | 0.504 | 0.509 |

### 3.2. Normal Distribution

Parameter ranges. The examples of 20-bit histograms for 8000 particles given in Supplementary Materials show their main feature: at $W > 1$, the proportion of small-sized particles increases due to the impossibility of negative values of the particle size, and at $W < 1$, it is located entirely on the positive part of the abscissa axis. Thus, the average dimensions of $\overline{L}_{1,2,3} \sim 40$–500 nm and the value of the normalized dispersion covering the point $W = 1$ were used in the calculations.

Dependences of $K_n$ on the correlation coefficients. Based on Figure 5, the following conclusions are made:

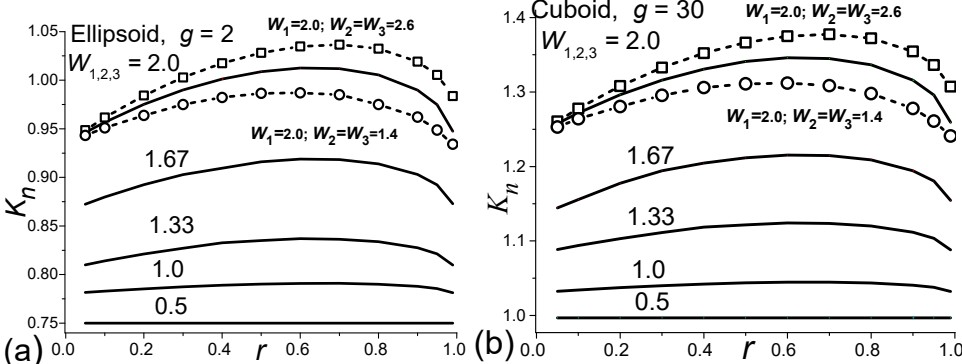

**Figure 5.** Dependences of $K_n$ on the correlation coefficient $r$ and the dispersion $W_1$ for ellipsoids (**a**) and bodies close to cuboids (**b**) using 20-bit normal distribution histograms. For the solid lines, $W_1 = W_2 = W_3$. For $W_1 = 2$, the calculations are shown for $W_2 = W_3 = 1.4$ (o) and $W_2 = W_3 = 2.6$ (□).

1. At a small $W < 0.5$–0.8, there is no dependence of $K_n$ on $r$ and $W$, as well as no dependence of the calculated values $\overline{D}_1\,(M_1)$ on the values of the distribution parameters along the second and third axes, as is observed in Figure 3 for the values of $r > 0.7$ of lognormal distributions. The values of $K_n$ calculated at small W, up to nearly 1, can be considered universal constants equal to $K_{g,0}$, with a small error (Table 1).

2. For large $W$ up to 2, the dependences along the 2nd and 3rd axes are significantly weak. In particular, as can be seen from Figure 5, for the curves with $W = 2$, the 30% changes in the values of dispersion lead to much smaller changes of less than 3% in $K_n$.

Thus, the calculations with equal parameters along all three axes can be considered accurate. With anisotropic deviations of up to 30% in the distribution parameters at large $W$, the error of 3% can occur.

Dependences of $K_n$ on dispersion $W_1$. Figure 6 shows the results of calculations that can also be directly used in the development of technology for electrode crystalline powders with normally distributed crystallite sizes to determine block structure [49], assess the potential of the developed technologies, and optimize it [14].

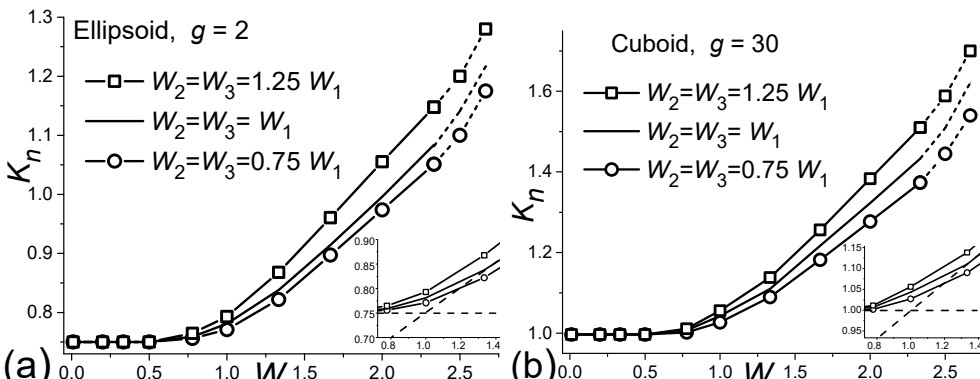

**Figure 6.** Dependence of $K_n$ on normal dispersions $W_1$ at $r = 0.5$ for ellipsoids (**a**) and cuboids (**b**) using 20-bit histograms, respectively, and dotted ends using 30-bit histograms. The figures show the relationships between the dispersions. The insets for the ratios, $W_1 = W_2 = W_3$, show approximations by dotted lines on an enlarged scale.

As can be seen from Figure 6, the solid lines for $W_1 = W_2 = W_3$ can be approximated by two straight lines in the region, $W_1 < 2.4$, that intersect at the point, $W = 1$. Analytically, the situation can be described by the so-called piecewise-continuous function [52]:

$$K_n = K_{g,0} \begin{cases} 1 & ; W < 1 \\ \frac{2}{3} + \frac{1}{3}W & ; W = 1 \div 2 \end{cases} \tag{13}$$

here the values of the constants, $K_{g,0}$, are the same as in Table 1.

Obviously, the rather simple Equations (12) and (13) can be obtained as a result of analytical transformations within the framework of the corresponding approximations. However, a simpler way was to carry out numerical calculations with subsequent approximation of the calculated curves.

## 4. Sequence of Practical Application to Optimize the Technology

The use of $K_{ni}$ coefficients for studying real powder crystallites and optimizing their technology can be carried out in the following sequence of stages:

1. According to [49], it is necessary (i) to determine the X-ray diffraction size $\overline{D}_i(M_i)$ using measurements and fairly reliable MAUD refinement software, (ii) to determine particle distribution of transverse $L_s$ and longitudinal $L_b$ sizes in their TEM images, (iii) to decompose into three marginal distributions of linear sizes $L_{[hkl]}$ along three crystallographic directions, using the calculated values of the coefficients $K_{ni}$, as well as information about their relative value $\overline{D}_i(M_i)$, for example, as they increase. This approach also allows the quantitive estimation of the percentage of composite particles, which consist of several crystallites and contain small- and large-angle boundaries.

2. Thus, we obtain 3D marginal distributions of linear crystallite sizes $\overline{L}_i$. Next, it is necessary to determine the parameter $\overline{\Lambda}$ matrix of correlation moments of the 3D lognormal (9) or normal (10) functions. To do this, according to [14], we use correlators between transverse $L_s$ and longitudinal $L_b$ dimensions as trial ones and, through inverse calculations, adjust them with the Mathematica 12 program to obtain marginal distributions using a trial 3D N-bit matrix obtained by discretizing the function $f_{LM}(\overline{L})$ (9) or $f_N(\overline{L})$ (10).

3. Thus, having obtained all the parameters of the 3D crystallite size distribution function, it is possible to use it to describe the physical parameters of crystallite powders, in particular, by using the example of electrode powder electrochemical properties and

their cathode rate capability [14]. Directly for 1D LiFePO$_4$ diffusion, it is necessary to average the capacitance over crystallite column length along [010] and then over all powder crystallites.

4. The task of optimizing the powder and improving its properties can be described as follows, particularly for electrode powders, dividing it into two subtasks:

   a. Achieving a large rate capability (and capacity) at big times, or increasing the rate capability at small times;

   b. Decreasing electrical relaxation time to increase the rate capability at small times, which is not directly solved through the technology of particles, but is necessary by improving the quality of their coatings [27].

## 5. Conclusions

The BWA (Bertaut–Warren–Averbach) method is an important tool for studying the structure of crystalline powders, namely, for relating the anisotropic X-ray diffraction sizes of crystallites to their anisotropic linear sizes (1). First, the lengths of the columns are averaged over the volume of a typical crystallite. Here, the anisotropic shape of the crystallite is important. Second, the lengths of the columns are averaged over all crystallites of the powder under study. Here, the distribution of crystallites over their anisotropic linear dimensions is important. A sign of these averaging is the presence of a relation for the dimensions (length)$^4$/(length)$^3$. In particular, therefore, the result of averaging over a crystallite is divided by the volume of this crystallite. Using these principles of the BWA method, the following results have been obtained:

1. Numerical calculations have been carried out for the crystallites with orthorhombic lattice, such as LiFePO$_4$ and NMC. The shape of the crystallites was approximated with the Lamé's $g$-type superellipsoids, including cuboids at large $g > 20 \div 30$. The values of the dimensional parameters of the powders were close to the ones observed experimentally.

2. In the presence of anisotropy, Equations (12) and (13) can be normalized to the length of the crystallite along its $i$-th axis; the normalized (dimensionless) distribution dispersions can also be used for lognormal and normal distributions. The universality of the results is limited by the discreteness of the matrices, which is due to the discreteness of the experimental histograms of crystallite sizes in the powders under study.

3. For lognormal distributions, the coefficient $K_n$ depends weakly on the correlation parameter $r$ for $r < 0.7$, i.e., in the range of values observed in high-quality LiFePO$_4$ samples. The observed dependence at $r > 0.7$ is related to the discreteness of the histograms and to the effect of crystallite size anisotropy along other crystallographic directions.

4. For the lognormal size distribution of crystallites, the following universal analytical expression for $K_n$ (12) was found, which is applicable up to values of $W_1 \sim 0.6$. For larger values, it is necessary to increase the bit number of the histograms, which requires a 2–3-fold increase in the number of measured particles.

5. For normal distributions with small $W < 0.5$–$0.7$, there is no dependence of $K_n$ on $r$ and $W$, and the calculated values, $\overline{D}_1 (M_1)$, are independent of the values of the distribution parameters along the second and third axes. The calculated $K_n$ for small $W \leq 1$ can be considered as a universal constant.

6. For normal distributions at large $W \leq 2$, the dependences of $K_n$ on the parameters of other axes are significantly weak, in particular, for the curves with $W_1 = 2$, 30% changes in dispersions along the second and third axes lead to much smaller changes in $K_n$ of no more than 3%.

7. For normal distributions, the dependences of $K_n$ on $W$ for the ratios, $W_1 = W_2 = W_3$, can be approximated by two straight lines that intersect at the point of $W = 1$. Analytically, the situation can be described by the so-called piecewise continuous function (13).

8. For gamma and Poisson distributions, mathematically reliable ways to describe correlations between sizes along different coordinate axes are lacking. In some cases, these

distributions are visually close to the lognormal and normal functions, which can be used instead of gamma and Poisson.

9. An elliptical cylinder (bar) (shown in Figure 1c) can be described as a combination of an ellipsoid with $M_1$ columns along the $x_1$ axis and a cuboid with $M_2$ columns along $x_3$.

10. The use of the BWA method is carried out on the example of LiFePO$_4$ and is justified by the fact that this compound has been studied quite well, so the development stages and final conclusions can be varied. In the other case of crystalline powders with a non-orthorhombic lattice, it is necessary to take into account the angles between the axes [9]. In this case, it becomes possible to widely use the BWA method for studying various crystalline powders: other electrode and atmospheric contaminants [47], as well as nuclear fuel and materials [53,54].

**Supplementary Materials:** The following supporting information can be downloaded at https://www.mdpi.com/article/10.3390/math11183963/s1, Figure S1 about lognormal, normal, gamma and Weibull distributions, Figures S2 and S3 about checking and controlling the stages of numerical calculations, Figure S4 about the dependence on the number of digits N of the column length, Figure S5 about comparing curves with 20 and 30 histogram sizes.

**Author Contributions:** A.B., conceptualization, writing, modeling; O.K., data analysis, validation; M.F., software, calculations; I.K. and XRD conceptualization, data curation. All authors have read and agreed to the published version of the manuscript.

**Funding:** This research received no external funding.

**Data Availability Statement:** Not applicable.

**Acknowledgments:** We express our sincere gratitude to Roman Davydov for the useful comments on the multiplication of 3D N-bit matrices and for the advice on checking procedures of program block numerical calculations.

**Conflicts of Interest:** The authors declare no conflict of interest.

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
