# Peer review of "Using the BWA (Bertaut-Warren-Averbach) Method to Optimize Crystalline Powders Such as LiFePO4"

_mathematics, doi:10.3390/math11183963_

Round 1

Reviewer 1 Report

This paper is well written and well organized. I strongly recommend it for publication.

Author Response

Dear Reviewer # 1

Thank you very much for reading the entire article and highlighting the most important parts.

Alexander Bobyl

Reviewer 2 Report

The authors present a study related to using the BWA (Bertaut‐Warren‐Averbach) method to optimize the technology of crystalline powders such as LiFePO4.
The authors propose the study of the constant Kn in the analysis of the BWA method to reduce the error of the results obtained from the present form of equation 1.
The results presented by the authors solve a problem of the existing method. This was supported by the analysis of an orthorhombic lattice of crystallites LiFePO4. 

Therefore, I consider that the manuscript could be accepted in the present form.

Author Response

Dear Reviewer # 2

Thank you very much for reading the entire article and highlighting the most important parts.

Alexander Bobyl

Reviewer 3 Report

In this article, the authors present their numerical calculations for determining X-ray diffraction sizes of crystallites in powders such as LiFePO4 and NMC. The topic is interesting and worthy of publishing. Before accepting for publication, the reviewer has some questions and may need minor revisions of the current manuscript.

1.      How do the authors calculate the X-ray diffraction sizes of crystallites in powders?

2.      What are the practical applications of optimizing the technology of crystalline powders using the BWA method?

3.      If there are other factors that could be influencing the results, it may be difficult to determine the true cause-and-effect relationship between the variables being studied. Please suggest some useful solutions.

4.      What are the potential sources of bias in the study and how were they addressed?

5.      Please introduce a bit more about the guiding role of these computational studies on actual experiments. Looking forward to hearing more from the author.

In this article, the authors present their numerical calculations for determining X-ray diffraction sizes of crystallites in powders such as LiFePO4 and NMC. The topic is interesting and worthy of publishing. Before accepting for publication, the reviewer has some questions and may need minor revisions of the current manuscript.

1.      How do the authors calculate the X-ray diffraction sizes of crystallites in powders?

2.      What are the practical applications of optimizing the technology of crystalline powders using the BWA method?

3.      If there are other factors that could be influencing the results, it may be difficult to determine the true cause-and-effect relationship between the variables being studied. Please suggest some useful solutions.

4.      What are the potential sources of bias in the study and how were they addressed?

5.      Please introduce a bit more about the guiding role of these computational studies on actual experiments. Looking forward to hearing more from the author.

Author Response

Dear Reviewer # 3

Thank you very much for reading the entire article, highlighting the most important parts and comments.

Attached are the answers to the points of your comments.

On behalf of authors.

Alexander Bobyl
